# High capacity topological coding based on nested vortex knots and links

Ling-Jun Kong [1,3], Weixuan Zhang [1,3], Peng Li[2,3], Xuyue Guo[2], Jingfeng Zhang[1], Furong Zhang [1], Jianlin Zhao [2] & Xiangdong Zhang [1✉]

Optical knots and links have attracted great attention because of their exotic topological characteristics. Recent investigations have shown that the information encoding based on optical knots could possess robust features against external perturbations. However, as a superior coding scheme, it is also necessary to achieve a high capacity, which is hard to be fulfilled by existing knot-carriers owing to the limit number of associated topological invariants. Thus, how to realize the knot-based information coding with a high capacity is a key problem to be solved. Here, we create a type of nested vortex knot, and show that it can be used to fulfill the robust information coding with a high capacity assisted by a large number of intrinsic topological invariants. In experiments, we design and fabricate metasurface holograms to generate light fields sustaining different kinds of nested vortex links. Furthermore, we verify the feasibility of the high-capacity coding scheme based on those topological optical knots. Our work opens another way to realize the robust and high-capacity optical coding, which may have useful impacts on the field of information transfer and storage.

---

[1] Key Laboratory of advanced optoelectronic quantum architecture and measurements of Ministry of Education, Beijing Key Laboratory of Nanophotonics & Ultrafine Optoelectronic Systems, School of Physics, Beijing Institute of Technology, 100081 Beijing, China. [2] MOE Key Laboratory of Material Physics and Chemistry under Extraordinary Conditions, and Shaanxi Key Laboratory of Optical Information Technology, School of Physical Science and Technology, Northwestern Polytechnical University, Xi'an 710129, China. [3]These authors contributed equally: Ling-Jun Kong, Weixuan Zhang, Peng Li. ✉email: zhangxd@bit.edu.cn

Transfer and storage of information are very important in current society, and information encoding is the first step to be implemented in these processes. Information coding with good stability and a high capacity is always the pursuing goal. Because the high capacity enables the efficient transfer of information, good stability could ensure the transmitted information maintains high quality. To date, most attentions have been focused on improving the capacity of optical coding schemes by utilizing different degrees of freedom, such as the wavelength[1], the polarization[2, 3], the orbital angular momentum[4], and so on. However, there is relatively little research on constructing robust all-optical information coding. In this case, how to create a novel coding scheme, which sustains good stability and a high capacity at the same time, is still to be solved.

On the other hand, the study of topological light fields, which are robust against perturbations[5–15], brings a potential way to solve such a problem. As typical topological light fields in real space, optical knots and links have attracted great attention in recent years[12–15]. Knots and links are defined mathematically as closed curves in three-dimensional space associated with physical strings[16]. At present, they have been observed in many physical systems[10–14,17–31], for example, it has been demonstrated that optical vortex knots and links can be generated by spatial light modulators (SLM) and designed nanostructures[10–14,32,33]. Recent investigations have also shown that optical framed knots can be used in conjunction with the prime factorization to encode information[34]. In topology, two knots are equivalent if one can be transformed into the other via continuous deformation without cutting the lines or permitting the lines to pass through itself. In other words, even if a knot structure may be deformed and distorted due to external disturbances, its topological invariants (such as the number of half twists used in ref. [34]) will remain unchanged. Therefore, the coding scheme based on the framed knots has good stability against external perturbations[34].

In addition to the robustness against external perturbations, it is necessary to achieve high capacity in practical coding. However, the capacity of the topological coding scheme based on the above optical framed knots is very limited, because it is very difficult to obtain a large number of topological invariants used for coding information. The question is whether or how the high capacity can be obtained in the coding process with knotted and linked structures. If the problem can be solved, it means that the coding scheme with both robustness and high capacity can be realized, which will have an important impact on the information society.

In this work, we theoretically propose and experimentally create the nested vortex knots in light fields to construct a topological coding scheme with high capacity. In particular, the so-called nested vortex knots refer to the layer-by-layer structures with fractal-like geometries. Based on the nested vortex knot, we can break through the number limitation of topological invariants to reach a high-capacity coding scheme. Specifically, we establish the theoretical framework of nested knotted and linked topological structures. Then, the topological coding scheme based on these structures is proposed. Furthermore, we have designed and fabricated the metasurface holograms to generate topological light fields with nested vortex links, and experimentally verified the feasibility of a topological coding scheme with a high capacity.

## Results

### The theory of constructing nested vortex knots and links.
The theory of knot and link has been extensively studied in the past century. It has shown that the abstract function with knotted or linked zeroes could be constructed by devising complex functions on a periodic braid embedded in a cylinder[10,35]. In this braid representation, braids are composed of twined strands. The most typical products are the Hopf link, trefoil knot, figure-8 knot, and cinquefoil knot[10,12,20,21]. Now, we consider hierarchically periodic braids. They are layer-by-layer structures, which are called nested structures. As an example, a nested three-strand braid twisted three times is shown in Fig. 1a. It contains three strands marked by green, red and blue. In order to show it more clearly, an enlarged view of the starting end plane is shown in Fig. 1b. The braid with a radius of $r_1$ is called the first-generation braid. Then, each strand with a radius of $r_2$ contains a sub-braid (called the second-generation braid, which is composed of three twined sub-strands). Similarly, each sub-strand with a radius of $r_3$ contains a sub-sub-braid (the third-generation braid, composed of three twined sub-sub-strands). Keep going in this way, we can finally construct a general nested braid structure with a fractal-like geometry.

In a general nested knot or link, each braid in the $i$th generation contains $N_i$ strands, where each strand can be labeled by a vector $\mathbf{M}_i = (m_1, m_2, \dots, m_i)$ and recorded as $S_{\mathbf{M}_i}$. Here, $m_i$ is the label number of the $m_i$-th strand in the $i$th generation and $m_i = 1, 2, \dots,$ or $N_i$. For example, $S_{3,2}$ represents the second strand included in the third braid for the first generation. Mathematically, $S_{\mathbf{M}_i}$ in the $(x', y', h)$ coordinate system can be described as:

$$x'_{\mathbf{M}_i}(h) = \sum_{\ell=1}^{i} r_\ell \cos\left(w_{\mathbf{M}_\ell} h + \varphi_{\mathbf{M}_\ell}\right), \quad (1a)$$

$$y'_{\mathbf{M}_i}(h) = \sum_{\ell=1}^{i} r_\ell \cos\left(w_{\mathbf{M}_\ell} h + \varphi_{\mathbf{M}_\ell}\right), \quad (1b)$$

where $\varphi_{\mathbf{M}_\ell}$ represents the initial rotation angle, which determines the starting position of each strand at the starting end plane of the braid. For example, in Fig. 1b, we show the initial rotation angles of the three strands in the first-generation braid $\varphi_1$, $\varphi_2$, and $\varphi_3$, and the initial rotation angles of the three strands in the third-generation braid $\varphi_{2,2,1}$, $\varphi_{2,2,2}$, and $\varphi_{2,2,3}$. $w_{\mathbf{M}_\ell}$ is named winding number, which represents the number of turns of the strand around the braid centerline. By introducing the complex coordinates $(u, v)$ as $u = x' + iy'$ and $v = e^{ih}$, the strands can be expressed as roots of a complex polynomials given by:

$$q(u, v) = \prod_{m_1=1}^{m_1=N_1} \prod_{m_2=1}^{m_2=N_2} \cdots \prod_{m_n=1}^{m_n=N_n} \left(u - \sum_{i=1}^{n} r_i v^{w_{\mathbf{M}_i}} e^{i\varphi_{\mathbf{M}_i}}\right) \quad (2)$$

Then, based on a stereographic projection[10,34], the complex polynomials could be expressed in the $(x, y, z)$ coordinate system as $f(x, y, z)$, which contains nested knotted and linked zero lines. Detailed derivations of $f$ are offered in Note 1 of the Supplementary Material.

As for the case with three generations ($n = 3$) and each generation containing three strands ($N_i = 3$) with all winding numbers equaling to 1, zero lines of $f(x, y, z)$ are in the form of a knotted and linked structure, as shown in Fig. 1c. The number of strands in the first, second, and third generation are 3, $3 \times 3 = 9$, and $3 \times 3 \times 3 = 27$, respectively. The total number of strands is 39. If only two generations are considered, that is setting $N_1 = 3, N_2 = 2$, $w_{\mathbf{M}_1} = 1$, and $w_{\mathbf{M}_2} = 1$, the corresponding nested braiding and linked structures are presented in Fig. 1d, e, respectively. In this case, the total number of strands decreases to 9. The detailed derivation of its Milnor polynomial is given in Note 2 of the Supplementary Material. And, the two-generation nested structures can be regarded as three different Hopf links (colored in green, red, and blue, respectively) nested with each other. If only considering one generation ($N_1 = 3$ and $w_{m_1} = 1$), the nested structure degenerates into the $6_3^3$ link (shown in Fig. 1f, g),

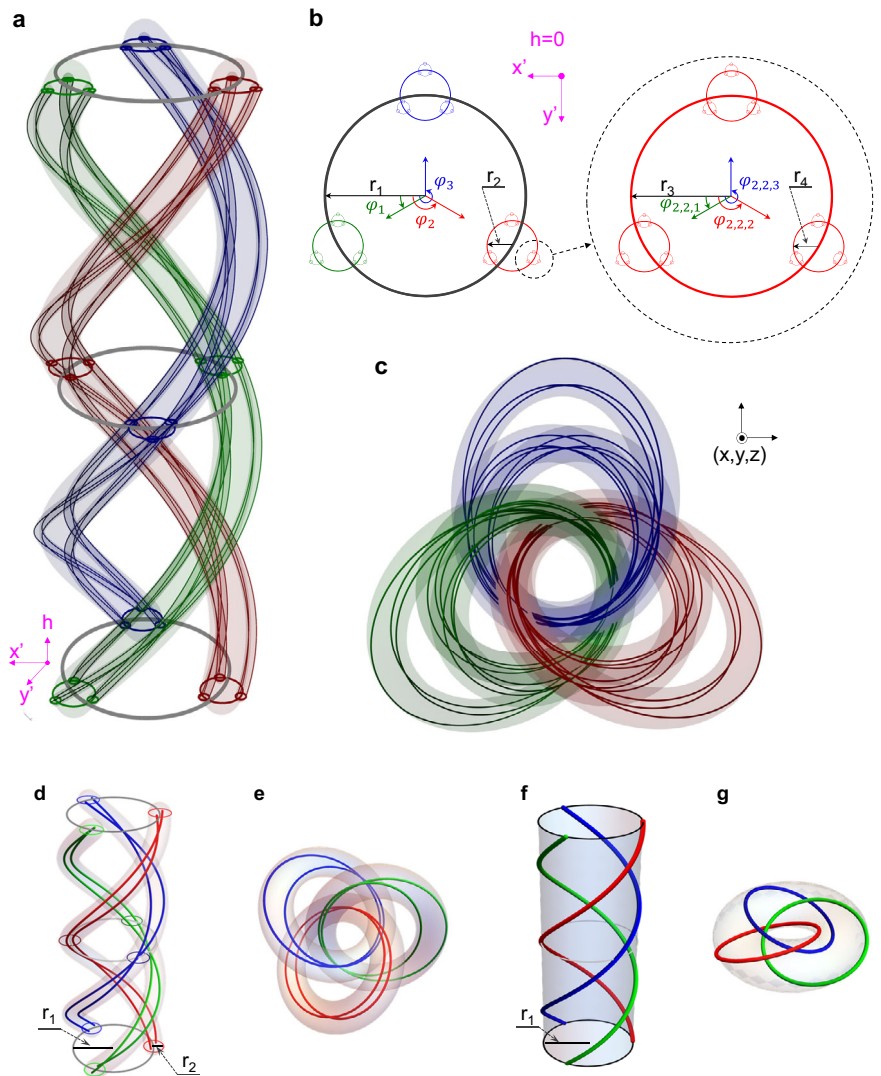

**Fig. 1 Theoretical construction of nested vortex knots and links. a** A nested three-strand braid twisted three times in the $(x', y', h)$ coordinate system, represented by Eq. (1) with $N_i = 3$, $w_{\mathbf{M}_i} = 1$. **b** Starting end plane of the nested three-strand braid in **a**. The centers of three circles with radius $r_2$ are on the arc of a circle with radius $r_1$. Similarly, the centers of three circles with radius $r_3$ are on the arc of circles with radius $r_2$. The position of each circle is related to its initial rotation angle. The initial rotation angles of three circles with radius $r_2$ ($r_4$) are marked with $\varphi_1$, $\varphi_2$, and $\varphi_3$ ($\varphi_{2,2,1}$, $\varphi_{2,2,2}$, and $\varphi_{2,2,3}$), respectively. **c** Nested link arising from the braid in **a** by using stereographic projection, which connects the $(x', y', h)$ and $(x, y, z)$ coordinates system. **d**, **e** The nested vortex link degenerated from the one shown in **a**, **c**, when we consider two generations and set the number of strands $N_1 = 3$ and $N_2 = 2$. **f**, **g** The nested vortex link degenerated from the one shown in **a**, **c**, when we consider only one generation and set $N_1 = 3$. The link shown in **g** contains three rings in the Hopf fibration and is marked as $6_3^3$ in Rolfsen's table.

which only contains three strands[36] (See Note 3 of the Supplementary Material for detailed derivation).

For a $n$-generation nested knotted and linked structure, the number of strands in the $i$-th generation can be calculated as $\prod_{j=1}^{i} N_j$. Therefore, the total number of strands is $A_N = \sum_{i=1}^{n}(\prod_{j=1}^{i} N_j)$, which is the sum of the number of strands in each generation. Each strand possesses a winding number. As in the traditional knot structure[34], these winding numbers are topologically protected and have good robustness, which can be used to code information and improve the anti-interference ability of the communication system.

**Coding scheme based on the nested knots and links**. According to the theoretical description in ref. [34], the coding scheme relies on a pair of numbers $(\alpha, \beta)$ where $\alpha$ is a positive integer, and $\beta$ is a number related to $\alpha$ and the associated topological structure. In our designed nested vortex knots, the number $\beta$ is given by

$$\beta = \prod_{i=1}^{i=n} \left[ \prod_{m_1=1}^{m_1=N_1} \prod_{m_2=1}^{m_2=N_2} \cdots \prod_{m_i=1}^{m_i=N_i} p_{\mathbf{M}_i}^{\left(\alpha^{w_{\mathbf{M}_i}}-W\right)} \right], \quad (3)$$

where $p_{\mathbf{M}_i}$ is a prime number assigned to the strand $S_{\mathbf{M}_i}$ and $W = \sum_{i=1}^{i=n} \left( \sum_{m_1=1}^{m_1=N_1} \sum_{m_2=1}^{m_2=N_2} \cdots \sum_{m_i=1}^{m_i=N_i} w_{\mathbf{M}_i} \right)$ represents the sum of all winding numbers. With these parameters, a natural number can be defined as $\mathbb{N}_{\alpha,\beta}(W) \overset{\text{def}}{=} \beta^{(\alpha^W)}$. Using Eq. (3), $\mathbb{N}_{\alpha,\beta}(W)$ can also be expressed by the prime factorization as

$$\mathbb{N}_{\alpha,\beta}(W) = \prod_{i=1}^{i=n} \left[ \prod_{m_1=1}^{m_1=N_1} \prod_{m_2=1}^{m_2=N_2} \cdots \prod_{m_i=1}^{m_i=N_i} p_{\mathbf{M}_i}^{\alpha^{w_{\mathbf{M}_i}}} \right], \quad (4)$$

which implies that the winding number set $\left\{ w_{\mathbf{M}_i} \right\}$ can be

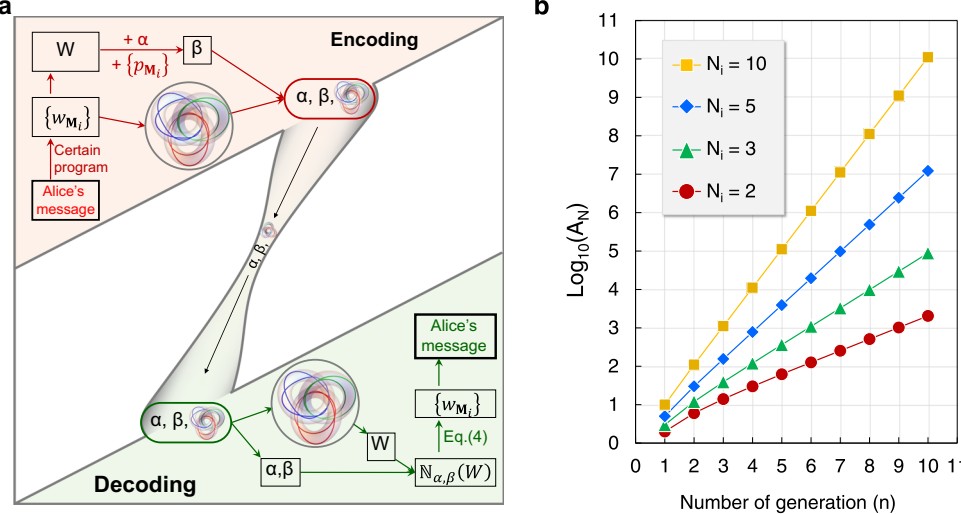

**Fig. 2 Schematic diagram of the communication protocol based on the nested knotted structure and its capacity. a** A communication protocol between Alice (sender) and Bob (receiver). In Alice's encoding part, Alice converts her message into a set of numbers $\{w_{M_i}\}$ by using a certain program. She calculates the $W$ with $\{w_{M_i}\}$, obtains $\beta$ with Eq. (3) by choosing a positive integer $\alpha$ and prime numbers $\{p_{M_i}\}$. On the other hand, she generates a nested knotted structure ($N_k$), whose set of winding numbers is exactly the $\{w_{M_i}\}$. Then, Alice sends the pair of numbers $(\alpha, \beta)$ and the $N_k$ to Bob in real time. In Bob's decoding part, after receiving these, Bob computes $\mathbb{N}_{\alpha,\beta}(W)$ with its definition, decodes the $\{w_{M_i}\}$ by using prime factorization as Eq. (4), and obtains Alice's message. **b** The relationship between the total number of the winding number ($A_N$) and the number of generations ($n$). Here, we set the number of strands in each braid to 2, 3, 5, or 10. That is, $N_i = 2, 3, 5,$ or 10. $A_N$ increases exponentially with the increase of $n$, and $A_N$ also increases rapidly with the increase of $N_i$.

obtained by the prime factorization to the natural number when the natural number $\mathbb{N}_{\alpha,\beta}$ is known. The detailed derivation from Eq. (3) to Eq. (4) and the prime factorization are given in Note 4 of Supplementary Material. Based on the definition and the prime factorization of the $\mathbb{N}_{\alpha,\beta}$, a communication protocol between Alice (sender) and Bob (receiver) can be established as follows. (I) As shown in the upper left corner of Fig. 2a, Alice converts the message into a set of numbers $\{w_{M_i}\}$ by using a certain program, and calculates the total values of winding numbers, $W$. For example, if Alice wants to send her name, she can first convert the word "Alice" into ASCII codes $\{w_{M_i}\} = \{65, 108, 105, 99, 101\}$, which will be regarded as the set of winding numbers, and calculates the total values of winding numbers $W = 478$. Then, Alice chooses a positive integer $\alpha$, assigns prime numbers $p_{M_i}$ to the winding number $w_{M_i}$, and calculates $\beta$ with Eq. (3). At the same time, a nested knotted structure ($N_k$) is generated, the set of winding numbers is exactly the $\{w_{M_i}\}$. (II) Alice sends the pair of numbers $(\alpha, \beta)$ and the $N_k$ to Bob in real time. (III) After receiving these, Bob computes $\mathbb{N}_{\alpha,\beta}(W)$ with its definition. As shown in the bottom right corner of Fig. 2a, Then, the $\{w_{M_i}\}$ can be decoded by using prime factorization as Eq. (4) and Alice's message can be obtained. It should be noted that there is a previously adopted convention, which clarifies how the $p_{M_i}$ is assigned to the $w_{M_i}$, for preventing $\{w_{M_i}\}$ to be an unordered set of integers.

In this coding scheme, where the topological invariants (winding numbers) are used as the information carriers, its capacity is determined by the total number of winding numbers $A_N$. While the $A_N$ is determined by the number of generations ($n$) and the number of strands contained in the braid in each generation ($N_i$). When setting $N_i = 2, 3, 5,$ or 10, the relationship between $A_N$ and $n$ is shown in Fig. 2b. Clearly, $A_N$ increases exponentially with the increase of $n$, and $A_N$ also increases rapidly with the increase of $N_i$. For $n = 10$ and $N_i = 10$, $A_N = 1.11 \times 10^{10}$. If we treat each strand in the nested knotted and

linked structure as 1-bit in our coding scheme, the capacity of information carried by the complete nested knotted and linked structure is $A_N$-bit. This means, a high-capacity coding scheme with topological robustness can be obtained by using the winding numbers of the nested knotted and linked structure as information carriers.

**Experimental demonstration.** Compared with the knotted or linked structures generated previously[10,12,20,30,34], the nested knotted and linked structures are more complicated. In this case, to create the nested knots at a fixed frequency, the designed hologram should contain more phase singularities in a limited transverse plane, which requires the hologram to have a higher resolution. What's more, the sensitivity and resolution of the detector are required to be high enough. In addition, a large number of phase singularities in the field at a single frequency could significantly interact with each other during the propagation in three-dimensional space, making it difficult to generate the complicated nested vortex knots. Fortunately, they can be generated experimentally by introducing multiple degrees of freedom and metasurface specially designed. Taking the link shown in Fig. 1g as an example, in the following, we will illustrate the generation process of embedding such a link into the designed light field. Considering this link consists of three rings, we can map the three rings of the nested link to vortex lines of light fields at three different wavelengths ($\lambda_1 = 532nm$, $\lambda_2 = 645nm$ and $\lambda_3 = 810nm$), respectively. The associated amplitude and phase distributions of the light fields at different wavelengths are shown in Fig. 3a (see Method for detailed derivation). Then, the required diffractive holograms for generating three rings are obtained by using the inverse sinc functional phase-only encoding technique[37,38] (Fig. 3b). Finally, the diffractive holograms are used to synthesize phaseonly metasurface hologram. More details on this process can be found in Note 5 in Supplementary Material.

The top view of the scanning electron microscope image of a fabricated metasurface is shown in Fig. 3c (scale bar, 1 μm). To

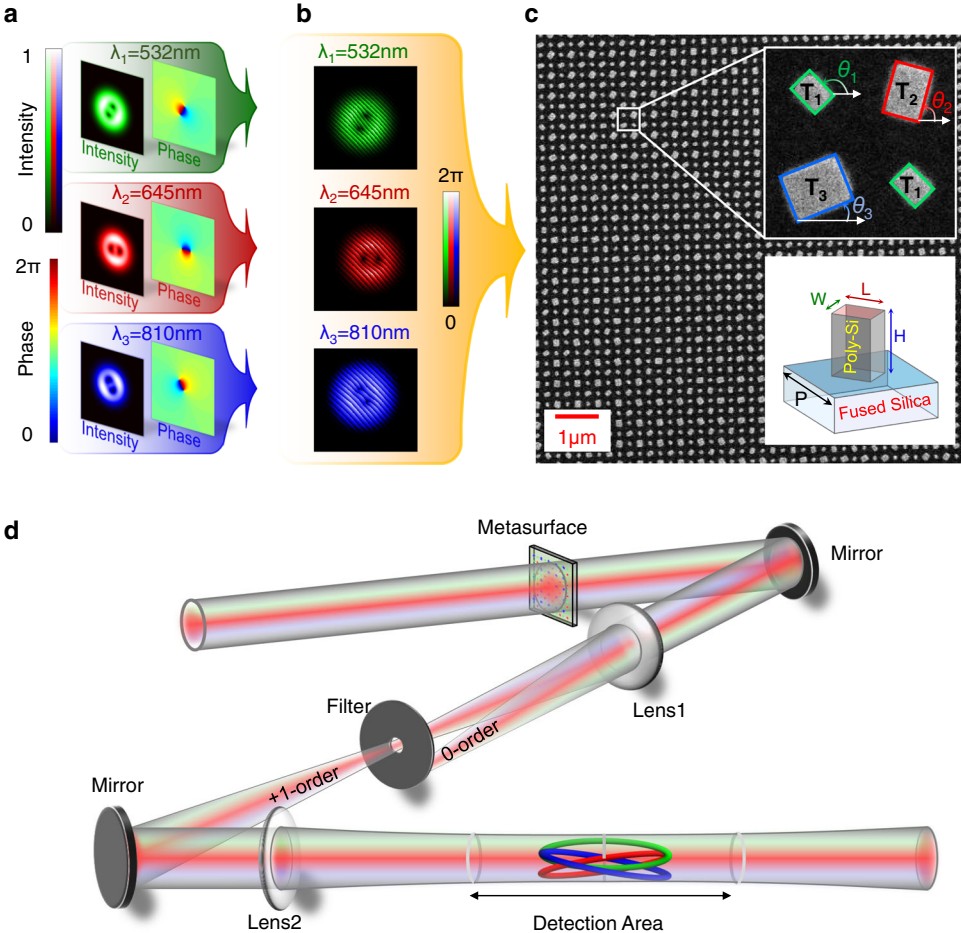

**Fig. 3 Experimental generation of nested vortex knots and links. a** Amplitude and phase profiles of three target fields. **b** Phase distributions sustaining knot vortex loops. A tilted phase grating is used to improve the signal-to-noise ratio by reducing the noise generated by the diffraction of the grating edges. The periods of the gratings should be adjusted appropriately to ensure that the first-order diffraction angles of the three fields with different wavelengths are the same. **c** Local scanning electron microscope (SEM) image of the designed metasurface for generating the nested link shown in Fig. 1g. The metasurface contains 600 × 600 square tetratomic macropixels with a side length of 500 nm. The up-right inset presents the enlarged view of a single tetratomic macropixel, which consists of four rectangle nanopillars with three types of geometries denoted as T₁, T₂, and T₃. Their orientation angles are marked with $\theta_1$, $\theta_2$, and $\theta_3$, respectively. The inset in the lower-right corner is a side view of a rectangle element, which consists of a rectangular polycrystalline silicon pillar on a fused silica substrate. L, W, H, and P represent the length, width, height, and period of the silicon pillar, respectively. **d** Experimental apparatus used to generate nested knots and links. The incident multi-wavelength light field consists of three different wavelengths (532, 645, and 810 nm). The metasurface is placed on the front focal plane of the 4 f system, which consists of lenses Lens1 and Lens2. A filter on the back focal plane of Lens1 filters out the light unwanted. The nested knots and links can be reconstructed by measuring the intensity distribution in the detection area with a CCD (not shown).

realize the functions of three diffractive holograms (shown in Fig. 3b) simultaneously, the metasurface constructed with tetratomic macropixels was designed[39–41]. The enlarged view of one tetratomic macropixel is shown in the inset in the upper-right corner. It is clearly shown that each tetratomic macropixel comprises three types of rectangle nanopillars with distinct geometries, which are denoted as T₁, T₂, and T₃, respectively. As shown in the inset in the lower-right corner, the metasurface consists of a fused silica substrate and a deposited polycrystalline silicon (Poly-Si) pillar. By optimizing the length (L), width (W), height (H), and period (P) of the rectangle nanopillar, we can make the nanopillar $T_i$ only respond to light fields with wavelengths $\lambda_i$ (i = 1, 2, or 3). The optimized geometric parameters are H = 300 nm, P = 250 nm, $(L, W)_1 = (93 \text{nm}, 70 \text{nm}), (L, W)_2 = (140 \text{nm}, 100 \text{nm})$, and $(L, W)_3 = (160 \text{nm}, 130 \text{nm})$ in our experiment. More details on the optimization process can be found in Note 5 in Supplementary Material. These nanopillars can be used to acquire a tunable geometric phase on the transmitted light based on the

spin-orbit interaction. The relationship between the geometric phase ($\phi_x$) and the orientation angle ($\theta_x$) of each type of nanopillar is $\phi_x = 2\theta_x$, with x = 1, 2, or 3. Guided by the above discussion, the three-phase profiles shown in Fig. 3b could be achieved by arranging the nanopillars in the metasurface plane according to the $\phi_x \sim \theta_x$ relationship. The detailed fabrication process of metasurface can be found in Method.

The experimental set-up used to observe nested knots and links with fabricated metasurface is shown in Fig. 3d. The multi-wavelength incident beam, which contains three monochromatic left-handed polarized light beams with $\lambda_1 = 532 \text{nm}$, $\lambda_2 = 645 \text{nm}$, and $\lambda_3 = 810 \text{nm}$, is manipulated to illuminate the metasurface vertically. The modulated beam, then, passes through a 4 f system, which consists of two lenses (Lens1 and Lens2) and a filter located at the back focal plane of the Lens1. In this case, only the first-order diffracted beam could be imaged on the back focal plane of the Lens2. Such an image plane is defined as the z = 0 plane, which corresponds to the central plane of light fields sustaining

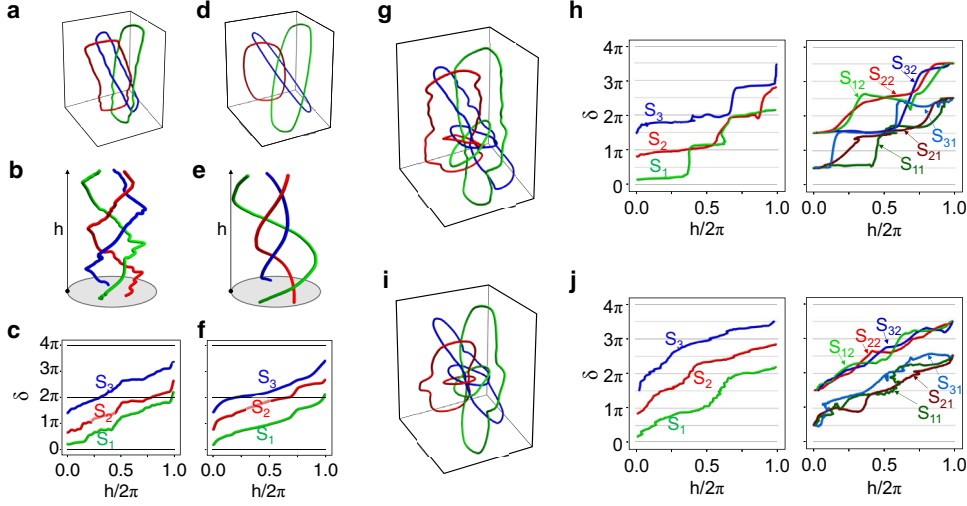

**Fig. 4 Optical nested vortex knotted and linked structures. a** Experimental results of the nested linked structure with one generation. **b** Braid versions of the nested link shown in **a**. **c** Twisting angles of three strands ($S_1$, $S_2$, and $S_3$) in the braid shown in **b**. **d–f** Theoretical results corresponding to the **a–c** based on the angular spectrum theory of the light field. **g** Experimental results of the nested linked structure with two generations. **h** Twisting angles of three strands in the first generation ($S_1$, $S_2$, and $S_3$) and six strands in the second generation ($S_{11}$, $S_{12}$, $S_{21}$, $S_{22}$, $S_{31}$, and $S_{32}$) contained in nested linked structure (shown in **g**). **i**, **j** are the theoretical results corresponding to the **g**, **h** based on the angular spectrum theory of light field, respectively.

nested vortex links. A CCD camera is used to measure the intensity profile of the light field. In front of the camera, there are three filters, each of which allows one wavelength to pass through during the measurement of intensity profiles. The CCD is placed on a translation stage (along the z-direction), allowing a full 3D scan of the transmitted beam.

From the measured intensity distributions, we can obtain the dark points (which are the intensity singularities) in each plane (see Note 6 in Supplementary Materials for more information). Connecting these dark points, an isolated vortex link is formed. The experimental results are shown in Fig. 4a, which contains three closed rings (link $6_3^3$). Its braid form is obtained by using stereographic projection and is shown in Fig. 4b. For comparison, the corresponding theoretical results for the isolated vortex link and braid form are given in Fig. 4d, e, respectively. It is seen clearly that the agreements between the theoretical and experimental results are very well. To extract the winding numbers of each strand, we need to introduce the parameter, twisting angle ($\delta$)[34]. The twisting angles ($\delta$) represents the azimuth angle, which is the function of the height ($h \in [0, 2\pi]$) of the strand. From Eq. (1), it can be expressed as $\delta_{M_\ell}(h) = w_{M_\ell} h + \varphi_{M_\ell} \cdot \delta_{M_\ell}(0) = \varphi_{M_\ell}$ represents the initial rotation angle. The winding numbers can be obtained with $w_{M_\ell} = [\delta_{M_\ell}(2\pi) - \delta_{M_\ell}(0)]/2\pi$. From Fig. 4b, e, we can obtain the twisting angles of three strands ($S_1$, $S_2$, and $S_3$) as a function of h/2$\pi$, which are shown in Fig. 4c, f for experimental and theoretical results, respectively. Both experimental and theoretical results show that the winding number for each strand is 1 and the total number is 3.

Next, we consider the nested link with two generations (shown in Fig. 1e). The measured nested link is plotted in Fig. 4g (details on the measured intensity distributions are given in Note 7 in Supplementary Materials). In this case, the twisting angle of each strand are shown in Fig. 4h. Three strands in the braid of the first generation are marked as $S_1$, $S_2$, and $S_3$, six strands in the three braids of the second generation are marked as $S_{11}$, $S_{12}$, $S_{21}$, $S_{22}$, $S_{31}$, and $S_{32}$. Because the winding number for each strand is 1, the total number is 9 in such a case. The corresponding theoretical results are plotted in Fig. 4i, j. The agreements between the experimental and theoretical results are observed again.

In addition, we would like to point out that the winding numbers in nested vortex knots and links are topological invariants. Here, we take the $6_3^3$ link as an example. By comparing the experimental results (in Fig. 4a) and the simulation results based on the angular spectrum theory (in Fig. 4d), it can be seen that the detailed shape of each ring constituting the $6_3^3$ link is different from each other due to the perturbations, such as deviation of the incident angle, the deviation of the waist position of the incident beam caused by the external environmental disturbance, the imperfection of the fabricated metasurface, and so on. However, the knotted structures are still the same, because the three rings are nested in the same way. Thus, the coding scheme based on our nested knotted structures has good stability against external perturbations.

## Discussion

In the above experiments, the nested links are only generated by utilizing the light field of three kinds of frequency. In such a case, the coding capacity is not very high. In fact, using the existing technology, more than a dozen or even dozens of frequencies can be used in wavelength division multiplexing or frequency comb technology[42–44]. When so many frequencies are used, the capacity in our coding scheme can reach a very high level. Of course, it is very difficult to generate topological light fields with so many frequencies. However, with the improvement of technology, we believe that the corresponding difficulties should be overcome.

At present, there are different kinds of optical devices used to generate the optical knotted and linked structures, such as metasurfaces[14,32], SLMs[12,20], and so on. Compared with the SLM, metasurface has many advantages, such as the ultra-small scale[14], the ultra-thin thickness, integrability[45], the large band span[39], the modulation of multi-degree of freedom of light field[39,45], and so on. On the other hand, metasurfaces are not as flexible as SLMs, where the hologram can be refreshed. This is the shortcoming in the practical application of our coding scheme based on metasurfaces. Fortunately, the research of reconfigurable metasurfaces has also made great progress in recent years[46,47], which may provide a possible way to make up for this shortcoming.

When the nested knotted and linked structures become more complicated, the number of intensity singularities in the intensity distribution will increase, and the distance between intensity singularities may become too small for the detector to distinguish them. In this case, an interferometer can be used to extract phase singularities[32] or polarization singularities[12,34] to assist in the reconstruction of the knotted and linked structures.

Recently, knots formed by three-dimensional polychromatic waves were explored[48,49], where position-dependent knots, defined by the trajectory of the electric field, are generated. It might be interesting, from both fundamental and application points of view, to combine the concepts of position-dependent knots and our nested knots together, which may increase the capacity further.

In summary, we have constructed another kind of vortex knotted and linked structures, nested vortex knots and links. Based on these structures, we have broken through the number of limitation of topological invariants in the existing knot and link structures and established a topological coding scheme with both good stability and high capacity. In the experiments, based on the designed metasurface, the nested linked structures with two generations have been generated and the feasibility of the coding scheme has been demonstrated. Our studies have paved the way for the realization of information coding with good stability and high capacity.

## Methods

**Sample fabrication.** The metasurfaces were fabricated based on the process of deposition, patterning, lift-off, and etching. At first, a 300 nm thick Poly-Si film was deposited on a 500 μm thick fused silica substrate by inductively coupled plasma-enhanced chemical vapor deposition (ICPECVD), and then a 100 nm thick Hydrogen silsesquioxane electron beam spin-on resist (HSQ, XR-1541) was spin-coated onto the Poly-Si film and baked on a hot plate at 100 °C for 2 min. Next, the desired structures were imprinted by using standard electron beam lithography (EBL, Nanobeam Limited, NB5) and subsequently developed in NMD-3 solution (concentration 2.38%) for 2 min. Finally, by using inductively coupled plasma etching (ICP, Oxford Instruments, Oxford Plasma Pro 100 Cobra300), the desired structures were transferred from resist to the Poly-Si film.

## Data availability

All data are displayed in the main text and Supplementary Information. The data that support the findings of this study are available from the corresponding author upon reasonable request.

## Code availability

The code that supports the plots within this paper are available from the corresponding author upon reasonable request.

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

## Acknowledgements
This work was supported by the National Key R & D Program of China under Grant No. 2017YFA0303800 and the National Natural Science Foundation of China (No.91850205).

## Author contributions
L.-J.K. finished experiment measurements with the help of Jf.Z. and F.Z. W.Z. finished the theoretical scheme with the help of L.-J.K. P.L. fabricated the sample with the help of X.G. and Jl.Z. L.-J.K., W.Z., and X.Z. wrote the manuscript. X.Z. initiated and designed this research project.

## Competing interests
The authors declare no competing interests.
