## [Peer review file · Nature Communications]

REVIEWER COMMENTS

Reviewer #1 (Remarks to the Author):

General Comments:

In this paper the authors propose to use nested vortex knots for high-capacity and robust information coding, which is supported by a large number of intrinsic topological invariants.

The paper is generally well written and the proposed approach seems to advance the state-of-the-art in this field. However, the authors should present more clearly the major differences and similarities in terms of contributions relatively to previous works, notably Ref. [30]. Besides this, the state-of-the-art review seems to be adequate and extensive.

Although the experimental setup and validation is convincing and support some of the claims of the authors, one aspect that should be better clarified is the current limitations of applicability, notably in terms of nested structures generation (in the transmission side) and in terms of detection (in the receiver side).

Finally, there are some concepts that should be better explained to help the general reader, as described in the detailed comments.

Provided that the authors clarify these aspects, it is the opinion of this reviewer that this paper could be published in this journal.

Detailed Comments:

Page 1:

Content: "Furthermore, we have designed and fabricated two metasurface holograms to generate light fields sustaining different nested vortex links, and experimentally verified the feasibility of the high-capacity coding scheme based on such topological optical knots."

Comment: Too much information in a single sentence. It may not be clear for the general reader what are metasurface holograms and nested vortex links at this stage.

Page 2:

Content: " However, to date, it is still very difficult to propose a coding scheme sustaining a good stability and a high capacity at the same time. "

Comment: The authors should say in which specific context this statement is true.

Page 2:

Content: "which is robustness against perturbations [1–11]"

Comment: Revise this sentence, e.g., "which is/are robust against perturbations [1–11]"

Page 2:

Content: "In this work, we theoretically propose and experimentally create the nested vortex knots in light fields to construct a topological coding scheme with high capacity. In particular, the so-called nested vortex knots refer to the layer-by-layer structures with fractal-like geometries."

Comment: Why are nested vortex knots more robust to external perturbations?

Page 4:

Content: "When we consider two generations and set $N_1 = 3$ and $N_2 = 2$ "

Comment: Consider revising this sentence as follows: "When we consider two generations and set the number of strands $N_1 = 3$ and $N_2 = 2$ ".

Page 4:

Content: "When we consider only one generation and set $N_1 = 3$, the nested vortex link shown in a(c) will degenerate into one shown in f(g), which is actually the 6^3_3 link."

Comment: Explain in the text the meaning and how its obtained each number in 6^3_3 .

Page 5:

Content: " Each strand possesses a winding number. As in the traditional knot structure [30], these winding numbers are topologically protected and have good robustness, which can be used to code information and improve the anti-interference ability of communication system. "

Comment: Justify why the winding numbers are “topologically protected and have good robustness” and explain which real conditions may lead to inaccurate detection of these numbers.

Page 6:

Content: "Alice converts the message into a set of number"

Comment: Alice converts the message into a set of numbers. Moreover, replace message -> message along the text and figures.

Page 6:

Content: "Alice converts the message into a set of number {...} by using a certain program, and calculates the total values of winding numbers, "

Comment: It would help the reader to have a particular example to illustrate this step.

Page 6:

Content: "At the same time, a nested knotted structure (Nk) is generated, the set of winding numbers is exactly the {...}."

Comment: What kind of physical limitations affect the generation/detection of “high” dimension nested knotted structures?

Page 6:

Content: "Because the nested knotted and linked structures have very good scalability, the coding scheme based on these structures can have a high capacity."

Comment: How do you define scalability in this context? The authors should clarify what is “high capacity” in this context, e.g., spectral efficiency in bit/Hz, storage capacity in bit/in², etc.

Page 7:

Content: "Alice converts her message into a set of number "

Comment: Alice converts her message into a set of numbers

Page 7:

Content: "obtains the Alice’s message"

Comment: obtains the Alice's message

Page 8:

Content: "They are difficult to be achieved directly by only using the phase singularity distribution."

Comment: This statement should be better explained.

Page 8:

Content: "Finally, the diffractive holograms are used to synthesize phase-only metasurface hologram."

Comment: Explain briefly how this is performed.

Page 10:

Content: "no shown"

Comment: not shown

Page 10:

Content: "which is the intensity singularities"

Comment: which are the intensity singularities

Page 10:

Content: "To extract the winding numbers of each strand, we need introduce the parameter, twisting angle"

Comment: To extract the winding numbers of each strand, we need to introduce the parameter, twisting angle

Page 11:

Content: "The corresponding experimental results are plotted in Figs 4i and 4j. "

Comment: The corresponding experimental results are plotted in Figs 4g and 4h.

Page 11:

Content: "we would like to point out that the winding numbers in nested vortex knots and links are topological invariants, which are robust against perturbations. "

Comment: How is the statement of "robustness against perturbations" supported by evidence in this paper?

Page 11:

Content: "Thus, the coding scheme based on them has good stability against external perturbations."

Comment: Clarify what kind of external perturbations are worth to consider.

Reviewer #2 (Remarks to the Author):

In their manuscript titled "Topological coding with a high capacity based on nested vortex knots and links", Kong et al. expand on recently developed methods to encode information in optical knots and links. Optical knots/links consist in three-dimensional trajectories of phase singularities which form non-trivial topological structures which can be classified in terms of topological invariants based on knot theory. These trajectories can be unfolded into braided structures, whose topological nature is determined by the way in which the individual strands are intertwined.

Here the Authors introduce a generalisation of these structures, in which each strand can consist into another "multi-stranded" braid (and the same procedure can be extended indefinitely to each of the new strands). These nested knot structures allow for an high capacity coding of information in terms of winding numbers associated with each strand, thus pointing towards important technical improvements of the scheme first introduced in Ref. [30]. Remarkably, the Authors show that an encoding of natural numbers based on prime number decomposition, similar to the one in Ref. [30], can be realised based only on winding numbers, which, being topological invariants, should ensure robustness under small perturbations.

In order to experimentally realise such structures one can rely of the mapping of knotted trajectories as zeros of Milnor polynomials, which can directly be encoded in optical fields (after multiplications by envelope functions ensuring finite energy). It is claimed in the manuscript that a direct encoding of these nested structures in a single monochromatic field is difficult. However the authors do not elaborate more on this point. My understanding is that is mainly a problem of the transverse resolution that would be necessary for encoding nested knots or links with an high number of generations. I would recommend a more detailed justification of this step, since it is fundamental for introducing the multiwavelength encoding. The latter consists in associating a generation in the nested knot with a different optical wavelength. In this case, the implementation of many generations does not require anymore extremely high spatial resolution (which might increase prohibitively with the generation number) but each generation can be associated to a different wavelength. As the authors point out, a relatively high number of generations can be in principle realised with modern optical frequency combs.

At the same time the use of multivawelenght sources requires the application of a different phase hologram for each wavelength, which may require an high number of resources. This issue is surpassed in the manuscript by employing a meta surface which consists of composite pixels where each sub pixel element can resonate with a different wavelength and impart an independent phase. This clever solution may be in principle expanded to more than 3 wavelengths. However, in light of generating knot/links for communication purposes, the possibility of efficiently reconfiguring the encoding device would be desired, a feature that is (to my knowledge) absent in metasurface technology.

Despite some of the above mentioned limits, which I would encourage the authors to discuss more in the conclusion section, I think the manuscript contains interesting insights and novelties and deserves to be accessible to the broad and interdisciplinary audience of Nature Communications, provided that some minor revisions are implemented in the manuscript.

In the following I report some suggestions:

- I struggled a bit understanding the definition of the M_i vector (although is correctly reported). The authors show an example ($S_{3,2}$) but I think it may be helpful to also include a short sentence specifying that " m_k refers to the k -th generation strand".

- The reference [23] in the sentence "According to the theoretical description in Ref. [23]" does not seem to report any form of prime number encoding... I think the Authors are actually referring to Ref. [30].

- As mentioned above, more detailed should be given about the reason why is difficult to implement nested knots in a monochromatic beam.

- Some comments about the necessity of "reconfigurable metasurfaces" for the practical use of nested knots in communications would help understanding the state-of-the-art and the future challenges of this line of research.

- The trajectory of phase singularities has been reconstructed by locating intensity zeros. This approach may become prohibitive for more complicated fields (where pairs of singularities may be so close to be undistinguishable). The author may comment on how this difficulty can be overcome either by interferometry (i.e. looking for dislocations in interference patterns which can be obtained letting part of the zeroth diffraction order be transmitted), or by realising polarisation knots.

- There is a missing reference for 'overhomogenization' in the supplementary material.

- As a possible additional insight, there are recent works (Phys. Rev. Research 2, 042045(R) (2020), Phys. Rev. Research 3, 033226 (2021)) exploring knots formed by three-dimensional polychromatic waves. In this scenario one can have position dependent knots, where the knot is a local object defined by the trajectory of the electric field. In this configuration one can think of associating different regions of the transverse plane (for example the focal plane of an high numerical aperture objective) with different generations of a nested knot?

Reviewer #3 (Remarks to the Author):

The manuscript describes an experimental realization of nested vortex knots in topological light fields as well as a theoretical framework for encoding information within such structures. The nested construct the authors have in mind seems to me related to what in knots theory named "satellite knots" or "cable knots", which are links embedded within knotted toruses. Here, such embeddings may go on forever, giving rise to such complicated fractal-like topologies.

The realization and the experimental details are way above my head and so I would like to focus in my review on the topological encoding scheme. The scheme is fairly standard and is taken from Ref. [30] therein. For some reason, however, the authors cite Ref. [23] as the originator of this scheme. Going through Ref. [23] there is no mentioning of anything of that sort. So to me, the encoding scheme here is an application of the method in Ref. [30] to the nested vortex braids. In this respect, I have found no flaws in the derivation.

My overall impression of the paper is that its strength is buried within the experimental part. It does seem to be a marked contribution in a line of works aimed at realizing such modes of structured light. But as I mentioned I leave the verdict on this aspect of the work to a more competent reviewer.

Responses to Reviewers

Response to Reviewer#1:

General comments:

In this paper the authors propose to use nested vortex knots for high-capacity and robust information coding, which is supported by a large number of intrinsic topological invariants. The paper is generally well written and the proposed approach seems to advance the state-of-the art in this field. However, the authors should present more clearly the major differences and similarities in terms of contributions relatively to previous works, notably Ref. [30]. Besides this, the state-of-the-art review seems to be adequate and extensive. Although the experimental setup and validation is convincing and support some of the claims of the authors, one aspect that should be better clarified is the current limitations of applicability, notably in terms of nested structures generation (in the transmission side) and in terms of detection (in the receiver side). Finally, there are some concepts that should be better explained to help the general reader, as described in the detailed comments. Provided that the authors clarify these aspects, it is the opinion of this reviewer that this paper could be published in this journal.

Authors' Response:

We would like to thank the referee for the careful review and positive remarks about our work. Following referee's suggestions, more descriptions and discussions on the differences and similarities in terms of contributions relatively to previous works have been added in page 2 of the revised manuscript. The discussion on current limitations about the applicability of our method has been added in the Discussion section. Moreover, some concepts are further illustrated in detail. In the following, we give point by point reply to the comments.

{Content in page 1: "Furthermore, we have designed and fabricated two metasurface holograms to generate light fields sustaining different nested vortex links, and experimentally verified the feasibility of the high-capacity coding scheme based on such topological optical knots."}

Comment #1: Too much information in a single sentence. It may not be clear for the general reader what are metasurface holograms and nested vortex links at this stage.

Authors' Response: We would like to thank the reviewer for the kind suggestion. This sentence has been rewritten in the revised manuscript as following:

"Here, we create a new-type of nested vortex knot, and show that it can be used to fulfill the robust information coding with a high capacity assisted by a large number of intrinsic topological invariants. In experiments, we design and fabricate metasurface holograms to generate light fields sustaining different kinds of nested vortex links. Furthermore, we verify the feasibility of the high-capacity coding scheme based on those topological optical knots."

{Content in page 2: " However, to date, it is still very difficult to propose a coding scheme sustaining a good stability and a high capacity at the same time. "}

Comment #2: The authors should say in which specific context this statement is true.

Authors' Response: We would like to thank the reviewer for the comment. In the revised manuscript, we have added the following discussion in page 2 to clarify the statement about advantages of the all-optical coding scheme based on nested knots: *“To date, most attentions have been focused on improving the capacity of optical coding schemes with utilizing different degrees of freedom, such as the wavelength (or frequency), polarization, orbital angular momentum, and so on. However, there is relatively little research on constructing the robust all-optical information coding. In this case, how to create a novel coding scheme, which sustains a good stability and a high capacity at the same time, is still to be solved.”*

{Content in page 2: "which is robustness against perturbations [1–11]}

Comment #3: Revise this sentence, e.g., “which is/are robust against perturbations [1–11]”

Authors' Response: We would like to thank the reviewer for the kind suggestion. In the revised manuscript, this sentence has been corrected.

{Content in page 2: "In this work, we theoretically propose and experimentally create the nested vortex knots in light fields to construct a topological coding scheme with high capacity. In particular, the so-called nested vortex knots refer to the layer-by-layer structures with fractal-like geometries."}

Comment #4: Why are nested vortex knots **more** robust to external perturbations?

Authors' Response: We would like to thank the reviewer for the comment. At first, we want to clarify the robustness of knot-based information coding. It is noted that the mathematical knots are knotted lines whose ends are joined together so that they cannot be undone. In topology, two knots are equivalent if one can be transformed into the other via the continuous deformation without cutting the lines or permitting the lines to pass through itself. In our study, we can see that the measured optical vortex knots (in Figure 4a of the main text), despite being deformed and distorted owing to the existence of many external perturbations (compared to simulation results in Figure 4d of the main text), would maintain topologically invariant. In this case, the knot-based information coding can still work well.

On the other hand, the topological invariant (winding numbers) used in our work is similar to that used in the previous work [Nat. Commun. 11, 5119 (2020)], making both coding schemes possess a good robustness. But, our designed nested vortex knots contain much more winding numbers belonging to different generations of nest knots. And, each winding number could be used to carry the specific information. Therefore, our nested vortex knots can be used to build a coding scheme with a higher capacity comparing with the previous work [Nat. Commun. 11, 5119 (2020)].

In the revised manuscript, we have added the following description in page 2 to further explain the robustness of the coding scheme based on the optical knotted structures: *“In topology, two knots are equivalent if one can be transformed into the other via the continuous deformation without cutting the lines or permitting the lines to pass through itself. In other words, even if a knot structure may be deformed and distorted due to the external disturbances, its topological invariants (such as the number of half twists used in Ref. [34]) will remain unchanged.”*

{Content in page 4: "When we consider two generations and set $N1 = 3$ and $N2 = 2$ "}

Comment #5: Consider revising this sentence as follows: "When we consider two generations and set the number of strands $N1 = 3$ and $N2 = 2$ ".

Authors' Response: We would like to thank the reviewer for the kind suggestion. This sentence has been rewritten in our revised manuscript.

{Content in page 4: "When we consider only one generation and set $N1 = 3$, the nested vortex link shown in a(c) will degenerate into one shown in f(g), which is actually the 6^3_3 link."}

Comment #6: Explain in the text the meaning and how its obtained each number in 6^3_3 .

Authors' Response: We would like to thank the reviewer for the kind suggestion. The name of this link comes from the Rolfsen's table. In the revised manuscript, we have added the following description in page 4 and added a Reference [36] in the main text to give more details about the 6^3_3 knot.

"If we consider only one generation and set $N1 = 3$, the nested vortex link shown in a (c) will degenerate into the one shown in f (g), which contains three rings in the Hopf fibration and is marked as 6^3_3 in Rolfsen's table."

{Content in page 5: " Each strand possesses a winding number. As in the traditional knot structure [30], these winding numbers are topologically protected and have good robustness, which can be used to code information and improve the anti-interference ability of communication system. "}

Comment #7: Justify why the winding numbers are "topologically protected and have good robustness" and explain which real conditions may lead to inaccurate detection of these numbers.

Authors' Response: We would like to thank the reviewer for the comment. The mathematical knots are knotted lines whose ends are joined together so that they cannot be undone. In topology, two knots are equivalent if one can be transformed into the other via the continuous deformation without cutting the lines or permitting the lines to pass through itself. In our study, we can see that the measured optical vortex knots (in Figure 4a), despite being deformed and distorted owing to the existence of many external perturbations (compared to simulation results in Figure 4d), would maintain topologically invariant. In this case, the knot-based information coding can still work well.

On the other hand, it should be emphasized that the *significant* external disturbances, including a large deviation of the incident angle, the deviation of the waist position of the incident beam caused by the external environmental disturbance, the imperfection of the fabricated metasurface sample, and the limit spatial resolution of measurements, may lead to inaccurate detection of vortex lines and make the measured winding numbers deviate from the designed one.

In the revised manuscript, we have added the following discussion in page 13 to further justify that the winding numbers are "topologically protected and have good robustness".

"In addition, we would like to point out that the winding numbers in nested vortex knots and links are topological invariants. Here, we take the 6_3^3 link as an example. By

comparing the experimental results (in Fig.4a) and the simulation results based on the angular spectrum theory (in Fig.4d), it can be seen that the detailed shape of each ring constituting the 6_3^3 link is different with each other due to the perturbations, including the deviation of the incident angle, the deviation of the waist position of the incident beam caused by the external environmental disturbance, the imperfection of the fabricated metasurface sample, and so on. However, the knotted structures are still the same, because three rings are nested in the same way. Thus, the coding scheme based on our nested knotted structures has a good stability against external perturbations.”

{Content in page 6: "Alice converts the message into a set of number"}

Comment #8: Alice converts the message into a set of numbers. Moreover, replace message -> message along the text and figures.

Authors' Response: We would like to thank the reviewer for the careful review of our work. Spelling errors have been corrected.

{Content in page 6: "Alice converts the message into a set of number {...} by using a certain program, and calculates the total values of winding numbers, "}

Comment #9: It would help the reader to have a particular example to illustrate this step.

Authors' Response: We would like to thank the reviewer for the kind suggestion. In our revised manuscript, a particular example (as discussed in the following) has been added in page 6.

“For example, if Alice want to send her name, she can firstly convert the word “Alice” into ASCII codes $\{w_{(M_i)}\}=\{65,108,105,99,101\}$, which will be regarded as the set of winding numbers, and then calculate the total values of winding numbers $W = 478$.”

{Content in page 6: "At the same time, a nested knotted structure (Nk) is generated, the set of winding numbers is exactly the {...}."}

Comment #10: What kind of physical limitations affect the generation/detection of “high” dimension nested knotted structures?

Authors' Response: We would like to thank the reviewer for the comment. In the generation of nested optical knots with many generations, the required incident light field should possess a sufficient large cross section, and the used modulation devices must have an extremely high spatial resolution. As for the detection of nested optical knots with many generations, it will require detection device with a larger detection area, a higher resolution, and a high sensitivity.

In the revised manuscript, we have added the following discussion in page 8 to further stress the physical limitations affect the generation/detection of “high” dimension nested knotted structures.

“Compared with the knotted or linked structures generated previously [10,12,20,30,34], the nested knotted and linked structures are more complicated. In this case, to create the nested knots at a fixed frequency, the designed hologram should contain more phase singularities in a limited transverse plane, which requires the hologram must have a higher resolution. And, the sensitivity and resolution of the detector should also be high enough. In addition, a large number of phase singularities in the field at a single frequency could

significantly interact with each other during the propagation in the three-dimensional space, making it difficult to generate the complicated nested vortex knots.”

{Content in page 6: "Because the nested knotted and linked structures have very good scalability, the coding scheme based on these structures can have a high capacity."}

Comment #11: How do you define scalability in this context? The authors should clarify what is “high capacity” in this context, e.g., spectral efficiency in bit/Hz, storage capacity in bit/in², etc.

Authors’ Response: We would like to thank the reviewer for the comment.

In principle, following the terms of the noisy-channel coding theorem [B. Saleem, “Channel capacity,” 2007; Jim Lesurf, “Signals look like noise!” *Information and Measurement*], the capacity of a given channel is the highest information rate (in units of information per unit time) that can be achieved with arbitrarily small error probability. In optical field, different degrees of freedom can be used as the carrier of information. In principle, the higher the dimension of the degree of freedom used, the higher the capacity of coding scheme. Therefore, the orbital angular momentum degree of freedom (which has infinite dimensions in principle) can be used to construct a coding scheme with higher capacity than the polarization degree of freedom (which has only two dimensions). Similarly, in the coding scheme with topological optical knots, where each topological invariant is used as the information carriers, the level of the capacity corresponds to the number of the topological invariants contained in the knot structure. Because the nested knotted and linked structures can be easily scaling from the one with less generations into the one with more generations, the coding scheme based on these structures can have a high capacity comparing with the previous work [Nat. Commun. 11, 5119 (2020)].

On the other hand, to quantitatively describe the information capacity, a complete communication system should be established [Nat. Commun. 5, 4876 (2014); Nat. Photon. 6, 488 (2012)]. Only in this case, the specific data of capacity could be given as XXX bit/Hz or (bit/s)/Hz. Here, similar to many previous works [Nat. Photon. 9, 822, (2020); Nat. Photon. 14, 102, (2020)], we focus on designing novel optical fields, which possess the ability to carry more information. The construction of a complete communication system based on nested vortex knots is remained in the future work.

To avoid being misunderstood, we rewrote the paragraph in page 6.

{Content in page 7: "Alice converts her message into a set of number "}

Comment #12: Alice converts her message into a set of numbers.

Authors’ Response: We would like to thank the reviewer for the careful review of our work. Spelling errors have been corrected.

{Content in page 7: "obtains the Alice’s message"}

Comment #13: obtains the Alice’s message

Authors’ Response: We would like to thank the reviewer for the careful review of our work. Spelling errors have been corrected.

{Content in page 8: "They are difficult to be achieved directly by only using the phase

singularity distribution."}

Comment #14: This statement should be better explained.

Authors' Response: We would like to thank the reviewer for the comment. More description (as follows) has been added in page 8 in our revised manuscript to further explain the difficulty of achieving directly by only using the phase singularity distribution.

Compared with the knotted or linked structures generated previously, the nested knotted and linked structures are more complicated. In this case, to create the nested knots at a fixed frequency, the designed hologram should contain more phase singularities in a limited transverse plane, which requires the hologram must have a higher resolution, and the sensitivity of the detector should also be high enough. In addition, a large number of phase singularities in a single frequency could significantly interact with each other during the propagation in real three-dimensional space, making it difficult to generate the complicated nested vortex knots.

{Content in page 8: "Finally, the diffractive holograms are used to synthesize phase-only metasurface hologram."}

Comment #15: Explain briefly how this is performed.

Authors' Response: We would like to thank the reviewer for the comment. In the revised manuscript, we have added the following discussion in page 7 in Supplementary Material to further describe the synthetic process.

"To embed the nested optical knots into the waist of a Gaussian beam, the diffractive holographic scheme based on the phase-only metasurface hologram can be used. It is proved that the phase-only holograms can be modified to control not only the phase structure of the diffracted beams but their intensity [New J. Phys. 2005, 7, 55]. The general process is summarized as follows. Firstly, the phase distribution of knotted vortex field at the $z=0$ plane should be added with a suitable blazed diffraction grating to construct the required phase distribution of the designed hologram. In this case, the first-order diffracted energy is angularly separated from the other orders. Then, the desired intensity of the knotted beam in the $z=0$ plane is applied as a multiplicative mask to the phase distribution of the hologram, acting as a selective beam attenuator imposing the necessary intensity distribution on the first-order diffracted beam."

{Content in page 10: "no shown"}

Comment #16: not shown

Authors' Response: We would like to thank the reviewer for the careful review of our work. Spelling errors have been corrected.

{Content in page 10: "which is the intensity singularities"}

Comment #17: which are the intensity singularities

Authors' Response: We would like to thank the reviewer for the careful review of our work. Spelling errors have been corrected.

{Content in page 10: "To extract the winding numbers of each strand, we need introduce the parameter, twisting angle"}

Comment #18: To extract the winding numbers of each strand, we need to introduce the parameter, twisting angle

Authors' Response: We would like to thank the reviewer for the careful review of our work. The word "to" has been added in our revised manuscript.

{Content in page 11: "The corresponding experimental results are plotted in Figs 4i and 4j."}

Comment #19: The corresponding experimental results are plotted in Figs 4g and 4h.

Authors' Response: We would like to thank the reviewer for the careful review of our work. This sentence has been revised in our revised manuscript.

{Content in page 11: "we would like to point out that the winding numbers in nested vortex knots and links are topological invariants, which are robust against perturbations."}

Comment #20: How is the statement of "robustness against perturbations" supported by evidence in this paper?

Authors' Response: We would like to thank the reviewer for the comment. The mathematical knots are knotted lines whose ends are joined together so that they cannot be undone. In topology, two knots are equivalent if one can be transformed into the other via the continuous deformation without cutting the lines or permitting the lines to pass through itself. In our study, we can see that the measured optical vortex knots, despite being deformed and distorted owing to the existence of many external perturbations, would maintain topologically invariant. In this case, the knot-based information coding can still work well. Here, we take the 6_3^3 link as an example. By comparing the simulation result based on the angular spectrum theory (in Fig.4d) and the experimental result (in Fig.4a), it can be seen that the detailed shape of each ring constituting the 6_3^3 link is different with each other. However, the knotted structures are still the same, because three rings are nested in the same way in space.

On the other hand, the robustness against perturbations of knotted structures is also shown in some previous works. In [Nat. Commun. 11, 5119 (2020)], there are some differences in the shape of the curve described in the theoretical results (as shown in Fig. 2c in main text) and the experimental results (as shown in Fig. 5a in main text), but the knot structures formed by these curves are the same knot, the trefoil knot. As long as the knotted structure has not changed, the number of half twists used as information carriers, will remain unchanged. This is why the coding scheme in [Nat. Commun. 11, 5119 (2020)] has topological protection.

In the revised manuscript, we have added the following discussion in page 13 to support the statement of "robustness against perturbations" by evidence in our work.

"In addition, we would like to point out that the winding numbers in nested vortex knots and links are topological invariants. Here, we take the 6_3^3 link as an example. By comparing the experimental results (in Fig.4a) and the simulation results based on the angular spectrum theory (in Fig.4d), it can be seen that the detailed shape of each ring constituting the 6_3^3 link is different with each other due to the perturbations, such as deviation of the incident angle, the deviation of the waist position of the incident beam caused by the external environmental disturbance, the imperfection of the fabricated metasurface sample, and so on. However, the knotted structures are still the same, because three rings are

nested in the same way in space. Thus, the coding scheme based on our nested knotted structures has good stability against external perturbations.”

{Content in page 11: "Thus, the coding scheme based on them has good stability against external perturbations."}

Comment #21: Clarify what kind of external perturbations are worth to consider.

Authors' Response: We would like to thank the reviewer for the comment. As we described in the previous response, the external perturbations include the deviations of the incident angle of the optical beam, the deviations of the waist position of the incident beam, the imperfections of the metasurface sample, and so on.

In the revised manuscript, we have added the following description in page 13 to point out the external perturbations.

“In addition, we would like to point out that the winding numbers in nested vortex knots and links are topological invariants. Here, we take the 6_3^3 link as an example. By comparing the experimental results (in Fig.4a) and the simulation results based on the angular spectrum theory (in Fig.4d), it can be seen that the detailed shape of each ring constituting the 6_3^3 link is different with each other due to the perturbations, such as deviation of the incident angle, the deviation of the waist position of the incident beam caused by the external environmental disturbance, the imperfection of the fabricated metasurface sample, and so on.....”

Response to Reviewer#2

General comments:

In their manuscript titled "Topological coding with a high capacity based on nested vortex knots and links", Kong et al. expand on recently developed methods to encode information in optical knots and links. Optical knots/links consist in three-dimensional trajectories of phase singularities which form non-trivial topological structures which can be classified in terms of topological invariants based on knot theory. These trajectories can be unfolded into braided structures, whose topological nature is determined by the way in which the individual strands are intertwined.

Here the Authors introduce a generalization of these structures, in which each strand can consist into another "multi-stranded" braid (and the same procedure can be extended indefinitely to each of the new strands). These nested knot structures allow for a high capacity coding of information in terms of winding numbers associated with each strand, thus pointing towards important technical improvements of the scheme first introduced in Ref. [30]. Remarkably, the Authors show that an encoding of natural numbers based on prime number decomposition, similar to the one in Ref. [30], can be realized based only on winding numbers, which, being topological invariants, should ensure robustness under small perturbations.

In order to experimentally realize such structures one can rely of the mapping of knotted trajectories as zeros of Milnor polynomials, which can directly be encoded in optical fields (after multiplications by envelope functions ensuring finite energy). It is

claimed in the manuscript that a direct encoding of these nested structures in a single monochromatic field is difficult. However the authors do not elaborate more on this point. My understanding is that is mainly a problem of the transverse resolution that would be necessary for encoding nested knots or links with an high number of generations. **I would recommend a more detailed justification of this step, since it is fundamental for introducing the multiwavelength encoding.** The latter consists in associating a generation in the nested knot with a different optical wavelength. In this case, the implementation of many generations does not require any more extremely high spatial resolution (which might increase prohibitively with the generation number) but each generation can be associated to a different wavelength. As the authors point out, a relatively high number of generations can be in principle realised with modern optical frequency combs. At the same time the use of multiwavelength sources requires the application of a different phase hologram for each wavelength, which may require an high number of resources. This issue is surpassed in the manuscript by employing a meta surface which consists of composite pixels where each sub pixel element can resonate with a different wavelength and impart an independent phase. This clever solution may be in principle expanded to more than 3 wavelengths. **However, in light of generating knot/links for communication purposes, the possibility of efficiently reconfiguring the encoding device would be desired, a feature that is (to my knowledge) absent in metasurface technology.**

Despite some of the above mentioned limits, which I would encourage the authors to discuss more in the conclusion section, I think the manuscript contains interesting insights and novelties and deserves to be accessible to the broad and interdisciplinary audience of Nature Communications, provided that some minor revisions are implemented in the manuscript.

Authors' Response: We would like to thank the reviewer for the careful review, positive remarks and valuable suggestions, which help to greatly improve the manuscript. We note that our previous manuscript does not contain a detailed discussion on the difficulty to implement nested knots in a monochromatic beam. Based on kind suggestions of the reviewer, in the revised manuscript, we have added the following detailed discussion to illustrate the problem for creating nested vortex knots at a fixed frequency: *“Compared with the knotted or linked structures generated previously [10,12,20,30,34], the nested knotted and linked structures are more complicated. In this case, to create the nested knots at a fixed frequency, the designed hologram should contain more phase singularities in a limited transverse plane, which requires the hologram must have a higher resolution, and the sensitivity and the resolution of the detector should also be high enough. In addition, a large number of phase singularities in a single frequency could significantly interact with each other during the propagation in real three-dimensional space, making it difficult to generate the complicated nested vortex knots”*

Moreover, the following discussion on the possibility of efficiently reconfiguring the encoding device have been added in our revised manuscript: *“At present, there are different kinds of optical devices used to generate the optical knotted and linked structures, such as metasurfaces [Adv. Opt. Mater. 7, 1900263 (2019); Laser Photonics Rev. 14, 1900366 (2020)], SLMs [New J. Phys. 7, 55 (2005); Nat. Phys. 14, 1079 (2018)], and so on. Compared with spatial light modulator, the metasurface has many advantages, such as the ultra-small*

scale [Adv. Opt. Mater. 7, 1900263 (2019)], the ultra-thin thickness, integrability [Adv. Mater. 32, 1805912 (2020)], the large band span [Adv. Mater. 7, 2103192 (2021)], the modulation of multi-degree of freedom of light field [Adv. Mater. 32, 1805912 (2020); Adv. Mater. 7, 2103192 (2021)], and so on. On the other hand, metasurfaces are not as flexible as SLMs, where the hologram can be refreshed. This is the shortcoming in the practical application of our coding scheme based on metasurfaces. Fortunately, the research of reconfigurable metasurfaces has also made a great progress in recent years [Nat. Photon. 10, 60 (2016); Nano Lett. 21, 8715, (2021)], which may provide a possible way to make up for this shortcoming”.

In the following, we give detailed responses to each review’s comment.

Comment #1: I struggled a bit understanding the definition of the M_i vector (although is correctly reported). The authors show an example (S_{3,2}) but I think it may be helpful to also include a short sentence specifying that “ m_k refers to the k-th generation strand”.

Authors’ Response: We would like to thank the reviewer for the kind suggestion. Relevant description has been added in page 4 in our revised manuscript.

Comment #2: The reference [23] in the sentence “According to the theoretical description in Ref. [23]” does not seem to report any form of prime number encoding... I think the Authors are actually referring to Ref. [30].

Authors’ Response: We would like to thank the reviewer for the careful review of our work. It is true that the reference [23] should be changed to Ref. [30] in our previous manuscript. The wrong serial number has been corrected in our revised manuscript.

Comment #3: As mentioned above, more detailed should be given about the reason why is difficult to implement nested knots in a monochromatic beam.

Authors’ Response: We would like to thank the reviewer for the comment. Based on kind suggestions of the reviewer, in the revised manuscript, we have added the following detailed discussion to illustrate the problem for creating nested vortex knots at a fixed frequency: “Compared with the knotted or linked structures generated previously [10,12,20,30,34], the nested knotted and linked structures are more complicated. In this case, to create the nested knots at a fixed frequency, the designed hologram should contain more phase singularities in a limited transverse plane, which requires the hologram must have a higher resolution, and the sensitivity and the resolution of the detector should also be high enough. In addition, a large number of phase singularities in a single frequency could significantly interact with each other during the propagation in real three-dimensional space, making it difficult to generate the complicated nested vortex knots.”

Comment #4: Some comments about the necessity of “reconfigurable metasurfaces” for the practical use of nested knots in communications would help understanding the state-of-the-art and the future challenges of this line of research.

Authors’ Response: We would like to thank the reviewer for the comment. According to this suggestion, the relevant discussion on “reconfigurable metasurfaces” (as follows) has been added to the Discussion section in our revised manuscript.

“At present, there are different kinds of optical devices used to generate the optical knotted and linked structures, such as metasurfaces [Adv. Opt. Mater. 7, 1900263 (2019); Laser Photonics Rev. 14, 1900366 (2020)], SLMs [New J. Phys. 7, 55 (2005); Nat. Phys. 14, 1079 (2018)], and so on. Compared with spatial light modulator, the metasurface has many advantages, such as the ultra-small scale [Adv. Opt. Mater. 7, 1900263 (2019)], the ultra-thin thickness, integrability [Adv. Mater. 32, 1805912 (2020)], the large band span [Adv. Mater. 7, 2103192 (2021)], the modulation of multi-degree of freedom of light field [Adv. Mater. 32, 1805912 (2020); Adv. Mater. 7, 2103192 (2021)], and so on. On the other hand, metasurface is not as flexible as SLMs, where the hologram can be refreshed. This is the shortcoming in the practical application of our coding scheme based on metasurfaces. Fortunately, the research of reconfigurable metasurfaces has also made a great progress in recent years [Nat. Photon. 10, 60 (2016); Nano Lett. 21, 8715, (2021)], which may provide a possible way to make up for this shortcoming”.

Comment #5: The trajectory of phase singularities has been reconstructed by locating intensity zeros. This approach may become prohibitive for more complicated fields (where pairs of singularities may be so close to be undistinguishable). The author may comment on how this difficulty can be overcome either by interferometry (i.e. looking for dislocations in interference patterns which can be obtained letting part of the zeroth diffraction order be transmitted), or by realising polarisation knots.

Authors' Response: We would like to thank the reviewer for the comment. According to your suggestion, the relevant discussion has been added to the Discussion section in our revised manuscript.

When the nested knotted and linked structures become more complicated, the number of intensity singularities in the intensity distribution will increase, and intensity singularities may become too close to be distinguished by the detector. In this case, an interferometer may be used to extract phase singularities [Laser Photonics Rev. 14, 1900366 (2020)] or polarization singularities [Nat. Phys. 14, 1079 (2018); Nat. Commun. 11, 5119 (2020)] to assist the reconstruction of knotted and linked structures.

Comment #6: There is a missing reference for 'overhomogenization' in the supplementary material.

Authors' Response: We would like to thank the reviewer for the careful review of our work. The reference has been added in our revised manuscript.

Comment #7: As a possible additional insight, there are recent works (Phys. Rev. Research 2, 042045(R) (2020), Phys. Rev. Research 3, 033226 (2021)) exploring knots formed by three-dimensional polychromatic waves. In this scenario one can have position dependent knots, where the knot is a local object defined by the trajectory of the electric field. In this configuration one can think of associating different regions of the transverse plane (for example the focal plane of a high numerical aperture objective) with different generations of a nested knot?

Authors' Response: We would like to thank the reviewer for the kind recommendation of the related paper. We have cited them as Refs. [48, 49] in our revised manuscript. This is a very

interesting idea, which has a good enlightening significance for our next work. Here, according to your suggestion, some discussions have been added in the Discussion section in our revised manuscript.

“Recently, knots formed by three-dimensional polychromatic waves were explored [Phys. Rev. Research 2, 042045(R) (2020); Phys. Rev. Research 3, 033226 (2021)], where position dependent knots, defined by the trajectory of the electric field, are generated. It might be interesting, from both fundamental and application points of view, to combine the concepts of position dependent knots and our nested knots together, which may increase the capacity further.”

Response to Reviewer#3

General comments:

The manuscript describes an experimental realization of nested vortex knots in topological light fields as well as a theoretical framework for encoding information within such structures. The nested construct the authors have in mind seems to me related to what in knots theory named “satellite knots” or “cable knots”, which are links embedded within knotted toruses. Here, such embeddings may go on forever, giving rise to such complicated fractal-like topologies.

Authors’ Response: We would like to thank the referee for the careful review of our manuscript. In the mathematical theory of knots, a satellite knot is a knot that contains an incompressible, non-boundary-parallel torus in its complement. Every knot is either hyperbolic, a torus, or a satellite knot. As the reviewer pointed out that the “satellite knots” embedded within knotted toruses, while in our nested knots the embeddings can go on forever, which give rise to such complicated fractal-like topologies. From this perspective, "satellite knots" can be regarded as a special case of our nested knots.

Comment #1: The realization and the experimental details are way above my head and so I would like to focus in my review on the topological encoding scheme. The scheme is fairly standard and is taken from Ref. [30] therein. For some reason, however, the authors cite Ref. [23] as the originator of this scheme. Going through Ref. [23] there is no mentioning of anything of that sort. So to me, the encoding scheme here is an application of the method in Ref. [30] to the nested vortex braids. In this respect, I have found no flaws in the derivation.

Authors’ Response: We would like to thank the referee for the comment. Your understanding is correct. The wrong serial number has been corrected in our revised manuscript.

Comment #2: My overall impression of the paper is that its strength is buried within the experimental part. It does seem to be a marked contribution in a line of works aimed at realizing such modes of structured light. But as I mentioned I leave the verdict on this aspect of the work to a more competent reviewer.

Authors' Response: We would like to thank the referee for the comment. As the reviewer said, our work is a marked contribution in a line of works aimed at realizing such modes of structured light in experiment. At the same time, our work is also innovative in theory. We propose a new kind of knotted structures, give an accurate mathematical general expression that can be used to control the design of knotted structure very well, expand the types of knot structures, and increases the value of knot structure in the application of information communication.

REVIEWERS' COMMENTS

Reviewer #1 (Remarks to the Author):

General Comments:

In this revised version of the manuscript, the authors addressed satisfactorily the concerns raised by the first version. Therefore, it is the opinion of this reviewer that this paper could be published in this journal with some minor corrections as described below.

Detailed Comments:

Page 6:

Content: "For example, if Alice want to send her name, she can firstly convert the word "Alice" into ASCII codes"

Comment: For example, if Alice wants to send her name, she can firstly convert the word "Alice" into ASCII codes

Page 7:

Content: "calculate the total values of winding numbers"

Comment: calculates the total values of winding numbers

Page 7:

Content: "Fig. 2"

Comment: Replace message by message in Figure 2a.

Page 8:

Content: "which requires the hologram must have a higher resolution,"

Comment: Revise this sentence.

Page 12:

Content: "Experimental results of the nested linked structure with two generation. "

Comment: Experimental results of the nested linked structure with two generations.

Reviewer #2 (Remarks to the Author):

In their revised version of the manuscript, the authors, in my opinion, correctly addressed the reviewers comments.

In particular, as they correctly observe, the fact that reconfigurable meta surfaces have been demonstrated implies that their approach, in principle, may become of practical interest for the use of communication platforms based on optical knots.

Hence I would recommend the publication of this work in Nature Communications.

Reviewer #3 (Remarks to the Author):

The authors have successfully addressed my concerns. I thus recommend accepting the manuscript in its present form.

Responses to Reviewers

- The Referees' comments are given in **blue**.
- Our replies are given in **black**.
- Modifications in the manuscript itself are highlighted in *Italics*.

Response to Reviewer#1:

General comments:

In this revised version the manuscript, the authors addressed satisfactorily the concerns raised by the first version. Therefore, it is the opinion of this reviewer that this paper could be published in this journal with some minor corrections as described below.

Authors' Reply:

We would like to thank the referee for the careful review and positive remarks about our work. Thanks for recommending the publication of revised manuscript. Some minor mistakes have been corrected following your suggestions in our revised manuscript.

{Content in page 6: "For example, if Alice want to send her name, she can firstly convert the word "Alice" into ASCII codes"}

Comment #1: For example, if Alice wants to send her name, she can firstly convert the word "Alice" into ASCII codes.

Authors' Reply: We would like to thank the reviewer for the kind suggestion. Spelling errors have been corrected.

{Content in page 7: "calculate the total values of winding numbers"}

Comment #2: calculates the total values of winding numbers.

Authors' Reply: We would like to thank the reviewer for the kind suggestion. Spelling errors have been corrected.

{Content in page 7: "Fig. 2"}

Comment #3: Replace massage by message in Figure 2a.

Authors' Reply: We would like to thank the reviewer for the good suggestion. Spelling errors in Figure 2a have been corrected.

{Content in page 8: "which requires the hologram must have a higher resolution,"}

Comment #3: Revise this sentence.

Authors' Reply: We would like to thank the reviewer for the kind suggestion. This sentence has been rewritten in our revised manuscript as following.

"In this case, to create the nested knots at a fixed frequency, the designed hologram should contain more phase singularities in a limited transverse plane, which requires the hologram to have a higher resolution. What's more, the sensitivity and resolution of the detector are required to be high enough."

{Content in page 12: "Experimental results of the nested linked structure with two generation. "}

Comment #3: Experimental results of the nested linked structure with two generations.

Authors' Reply: We would like to thank the reviewer for the kind suggestion. Spelling errors have been corrected.

Response to Reviewer#2

General comments:

In their revised version of the manuscript, the authors, in my opinion, correctly addressed the reviewers comments.

In particular, as they correctly observe, the fact that reconfigurable meta surfaces have been demonstrated implies that their approach, in principle, may become of practical interest for the use of communication platforms based on optical knots.

Hence I would recommend the publication of this work in Nature Communications.

Authors' Reply: We would like to thank the reviewer for recommending the publication of revised manuscript.

Response to Reviewer#3

General comments:

The authors have successfully addressed my concerns. I thus recommend accepting the manuscript in its present form.

Authors' Reply: We would like to thank the reviewer for recommending the publication of revised manuscript.